# Mediating Mindfulness-Based Interventions with Virtual Reality in Non-Clinical Populations: The State-of-the-Art

**DOI:** 10.3390/healthcare10071220

**Published:** 2022-06-29

**Authors:** Chiara Failla, Flavia Marino, Luca Bernardelli, Andrea Gaggioli, Germana Doria, Paola Chilà, Roberta Minutoli, Rita Mangano, Roberta Torrisi, Gennaro Tartarisco, Roberta Bruschetta, Francesco Arcuri, Antonio Cerasa, Giovanni Pioggia

**Affiliations:** 1Institute for Biomedical Research and Innovation (IRIB), National Research Council of Italy (CNR), 98164 Messina, Italy; chiara.failla@irib.cnr.it (C.F.); flavia.marino@irib.cnr.it (F.M.); germana.doria@istitutomarino.it (G.D.); paola.chila@irib.cnr.it (P.C.); roberta.minutoli@irib.cnr.it (R.M.); ritamangano1979@gmail.com (R.M.); roberta.torrisi87@gmail.com (R.T.); gennaro.tartarisco@irib.cnr.it (G.T.); roberta.bruschetta@irib.cnr.it (R.B.); 2Classical Linguistic Studies and Education Department, Kore University of Enna, 94100 Enna, Italy; 3Become-Hub, 20123 Milan, Italy; luca.bernardelli@become-hub.com; 4Research Center in Communication Psychology, Università Cattolica del Sacro Cuore, 20123 Milan, Italy; andrea.gaggioli@unicatt.it; 5ATN-P Lab, I.R.C.C.S. Istituto Auxologico Italiano, 20123 Milan, Italy; 6S’Anna Institute, 88900 Crotone, Italy; f.arcuri@istitutosantanna.it; 7Pharmacotechnology Documentation and Transfer Unit, Preclinical and Translational Pharmacology, Department of Pharmacy, Health and Nutritional Sciences, University of Calabria, 87036 Rende, Italy

**Keywords:** mindfulness, virtual reality, negative mood, non-clinical populations, physiological measures

## Abstract

Mindfulness is one of the most popular psychotherapeutic techniques that help to promote good mental and physical health. Combining mindfulness with immersive virtual reality (VR) has been proven to be especially effective for a wide range of mood disorders for which traditional mindfulness has proven valuable. However, the vast majority of immersive VR-enhanced mindfulness applications have focused on clinical settings, with little evidence on healthy subjects. This narrative review evaluates the real effectiveness of state-of-the-art mindfulness interventions mediated by VR systems in influencing mood and physiological status in non-clinical populations. Only studies with an RCT study design were considered. We conclude that most studies were characterized by one single meditation experience, which seemed sufficient to induce a significant reduction in negative mood states (anxiety, anger, depression, tension) combined with increased mindfulness skills. However, physiological correlates of mindfulness practices have scarcely been investigated. The application of VR-enhanced mindfulness-based interventions in non-clinical populations is in its infancy since most studies have several limitations, such as the poor employment of the RCT study design, the lack of physiological measurements (i.e., heart rate variability), as well as the high heterogeneity in demographical data, technological devices, and VR procedures. We thus concluded that before applying mindfulness interventions mediated by VR in clinical populations, more robust and reliable methodological procedures need to be defined.

## 1. The Healing Power of Mindfulness

Mindfulness has been defined as *“the act of consciously focusing the mind in the present moment without judgment and without attachment to the moment”* [1]. This practice refers to the self-regulation of the focus on one’s experiences in the present moment with curiosity, openness, and acceptance [2]. Mindfulness practice has extensively and successfully been applied for the treatment of mood disorders in several clinical domains, including neurological patients, such as drug-resistant epilepsy [3], multiple sclerosis [4], and Parkinson’s disease [5], as well as in other clinical disorders such as cardiovascular diseases [6]. Generally, all these studies confirm that mindfulness practice led to a decrease in depressive/anxiety symptoms and rumination [7] as well as an increase in positive affects [8,9]. Benefits extend beyond psychological realms, also affecting immune functioning [10], working memory, sustained attention [7,9], and regulation of homeostasis in the hypothalamic–pituitary–adrenal axis [11].

Among the various different mindfulness-related practices, mindfulness-based intervention (MBI) seems efficient and easily applicable for reducing psychological stress in the management of various physical and mental health conditions [12,13]. MBI generally refers to practices that cultivate awareness and require paying attention to the present moment [14], thereby improving metacognitive awareness and allowing participants to shift their perspective (“reperceiving”) by reducing emotional reactivity [15]. Mindfulness-based stress reduction (MBSR) [16] is another well-known practice, generally used on the general population to increase the health of non-patients via “wellness” programs [17]. Research in non-clinical populations indicates that MBSR helps decrease depressive symptoms and rumination [7] as well as increase positive affect [8,9]. Lastly, mindfulness meditation (MM) is widely used and conceptualized as nonjudgmental attention to the present sensorial moment and mental experience [18,19,20,21]. MM interventions are effective in treating various types of physical and psychological problems observed in different clinical and non-clinical settings [22,23,24,25].

## 2. Translating Mindfulness Interventions into Virtual Environments

The efficacy of mindfulness practice is strongly influenced by the psychotherapist’s skills. However, it is possible to standardize mindfulness-related procedures using tools that simulate multisensorial perception or attention. Virtual reality (VR) has recently been proposed as an intermediate interface that makes learning mindfulness practices easier for patients with emotional dysregulation [26].

VR enables users to experience computer-generated environments within controlled experimental conditions. This technology was often used for mental health treatment in the 1990s [27]. The transition of VR into the psychological setting occurred in the early 2000s, basically to promote exposure-based therapy for anxiety disorders [28]. This kind of therapy is based on the theory of emotional processing, which postulates that fear memories are structures that contain information regarding fear stimuli, responses, and meaning. As such, the goal of the intervention is to activate and modify these structures of fear by presenting new incompatible information and facilitating emotional processing. Immersive VR systems exploit realistic 3D graphics, stereoscopic viewing, and head tracking to create interactive, first-person experiences that can be more ecologically valid than traditional, non-interactive experimental stimuli and produce users’ physiological responses that are consistent with real-world experiences [29,30,31]. VR-based techniques are ideal for exposure therapy, as the sense of presence experienced in VR enables the patients to be immersed in the feared environment, which is adapted to match specific aspects of their fear structures in order to activate and modify these structures. Most of the research on VR-related treatment has therefore been conducted for anxiety disorders, and the results suggest that behavioral-related practices mediated by VR have similar efficacy to traditional exposure interventions (face-to-face), showing stable benefits over time [32].

Recently, VR has also been proposed as a mediator to support mindfulness-related interventions [33,34]. VR technologies may pragmatically address the challenges related to environmental distraction by providing an immersive, engaging, and controlled visual and auditory sandbox in which the participant can rehearse mindfulness skills, shifting attention away from the real-world environment [35,36,37].

However, different VR technologies and experimental setups are used during this kind of behavioral intervention, and there are no established guidelines to date or comparisons of methods.

## 3. Virtual Reality Procedures Used to Provide Mindfulness Practices in Non-Clinical Populations

In order to highlight the state-of-the-art in immersive VR-enhanced mindfulness applications, this review discusses the current technology and methodology used with non-clinical individuals. Despite the large number of papers demonstrating the effectiveness of mindfulness practices in clinical settings and the VR-related applications in psychotherapist settings, few studies have assessed the best practices for combining these two approaches. We thus present an overview of the commercially available instruments and address the instrumental and methodological aspects in order to establish the strengths and weaknesses of this new field of study.

## 4. Methods

This review was planned and conducted in accordance with the Preferred Reporting Items for Systematic Reviews and Meta-Analyses (PRISMA) guidelines (Figure 1). Articles published between 2002 and January 2022 were reported using electronic bibliographic databases such as PubMed, Scopus, and Google Scholar. To improve the search strategy, keywords including “text words” and MeSH were used. The search terms were concatenated in an advanced query using Boolean operators as follows: “Mindfulness” AND “virtual reality” AND (“non-clinical population” OR “healthy controls” OR “healthy”). After the initial web search, duplicate items among databases were removed. To reduce the risk of bias, two authors (C.F. and A.C.) independently screened paper abstracts and titles and analyzed the full papers that met the inclusion criteria, as suggested by the PRISMA guidelines. The reference lists of examined full-text papers were also scrutinized for additional relevant publications.

Criteria for including or excluding papers were determined a priori. Papers were considered for inclusion only if they: (a) were written in full-text English language in a peer review journal; (b) included non-clinical patients; (c) used randomized controlled trials (RCTs); and (d) used mindfulness practices combined with VR systems. Articles were excluded if they (a) considered other meditative practices such as transcendental meditation, autogenic training, or yoga; (b) were unpublished dissertations, book chapters, and conference papers. We further remove all articles not directly interested in evaluating the effects of mindfulness in improving psychological outcomes (i.e., anxiety, depression, awareness).

The data collected from each article were categorized as information on the first author and year of publication, the VR system, the size of cohorts, the modalities of intervention, the experimental procedures, the outcomes (physical and psychological), and the main results.

## 5. Results

Figure 1 reported the four phases—identification, screening, eligibility, and inclusion—of the process for the selection of the studies in this qualitative review. N = 104 remained after the first reviewing process. In the second phase, 86 were excluded because they did not fulfill the inclusion criteria. In the eligibility phase, we removed n° 11 articles interested in assessing the effect of immersive VR-enhanced mindfulness applications in other psychological domains (i.e., gaming). Finally, seven RCT articles were included in this review, describing experiments where healthy individuals underwent mindfulness-related intervention mediated by VR to improve psychological and/or physiological status. Studies without a control group or assessing VR-mediated mindfulness applications in clinical settings (for instance: [35,38,39,40]) were excluded. The trial size ranged between 26 and 100 participants. The mean age of participants recruited ranged between 18 and 65 years. In total, 387 participants were enrolled across seven trials, and six out of seven studies reported the gender of participants (woman vs. man). Among the studies, six recruited both male and female participants. All studies (N = 7) examined a control group.

In this section, we summarize the main evidence provided by the seven studies that were in line with the inclusion/exclusion criteria. In order to facilitate comparisons, Table 1 summarizes the relevant findings.

Tarrant et al., 2022

This study examined the effectiveness of a VR meditation experience combined with neurofeedback interfacing (experimental condition) for improving mood status in health workers (n° 50, 43 females; mean age = 42.1 years). Individuals enrolled in the control condition underwent a standard audio-only guided meditation (n° 50, 48 females; mean age = 40.9 years). The VR meditation experience also included the recording of biofeedback data from an EEG consisting of only two active electrodes placed over the frontal cortex.

Mindfulness practice consisted of an audio track that guided individuals in a body-scan/relaxation meditation experience. This is a standard procedure embedded in the Healium VR Experience called “beach relaxation”. The audio track was led by a female voice, which directed the subjects’ attention to specific parts of the body, inviting relaxation in each part. The experimental group wore an Oculus Brainlink Lite headband, which combined immersive experience simultaneously with the users’ brain wave activity provided via an EEG headband. The goal was to keep high beta activity below the threshold, providing the user with biofeedback information about the state of inner calm [46,48]. The control group listened to the same audio track employed for the experimental group without a VR interface.

After one single meditation experience, the experimental group showed significant decreases in scales measuring Fatigue, Confusion, and Vigor and an increase in Calmness and Happiness with respect to the control group.

Waller et al., 2021

This study assessed the different impacts of the instructor-guided meditation approach when delivered face-to-face with respect to pre-recorded 360° videos viewed through a 2D monitor or VR system. No biofeedback or electrophysiological data were collected.

A total of 82 university students (50 females) were randomly assigned to complete two similar guided meditation sessions in counterbalanced order. All participants completed a pre-recorded VR-based mindfulness guided meditation, and each participant completed a non-VR version of the same guided meditation, performed either as a live face-to-face session or by watching the pre-recorded 360° video via a laptop monitor and headphones. All guided meditation formats lasted approximately 5 min with script-based, trauma-informed meditation instructions that were kept consistent throughout each session.

Despite the authors not performing a clear separation between conditions and not evaluating the demographic and psychological characteristics of the experimental and control groups at baseline, the results showed that the VR meditation experiences were associated with greater relaxation and less distractibility of the breathing process compared to the non-VR format.

Tarrant et al., 2018

In this previous study, Tarrant and colleagues had used a different experimental setup to confirm whether immersive VR-enhanced mindfulness applications could reduce anxiety levels. With respect to the work published in 2022, this paper did not employ an EEG as a neurofeedback interfacing but as an instrument to evaluate neurophysiological changes during the meditation experience.

The study sample was divided into an experimental group that underwent one single VR-mediated mindfulness practice (n = 14; 11 females, mean age: 46.2) and a control group engaged in a standard condition of rest with eyes open (n = 12; 9 females, mean age: 48.1)

Each group underwent three time points of measurement, each consisting of 5 min of an eyes-closed EEG recording, and filled out an anxiety state questionnaire. EEG recordings were interleaved by an initial 5 min period of eyes-open rest for both groups, followed by either a brief 5 min VR meditation intervention or another period of 5 min eyes-open rest. The mindfulness experience lasted 5 min and 41 s where individuals were guided by a woman’s voice through an awareness meditation, directing attention to the elements of the environment and asking them to connect with what they were looking at whilst being immersed in 3D natural landscapes.

Despite both groups reporting decreasing anxiety during the treatments, electrophysiological activity was significantly reduced only in the experimental group, which demonstrated global and regional decreases in Beta activity during the VR meditation.

Yildmir et al., 2021

The authors evaluated whether a mindfulness intervention mediated by VR could be more effective in inducing a greater state of awareness than an intervention based on simple audio content.

A total of 45 undergraduate students (27 females) were divided into three groups with 15 students in each group: (a) experimental group with the VR-based mindfulness intervention; (b) group performing an audio-based mindfulness intervention; and (c) control group (listening to an audiobook).

The meditation practice for the two experimental groups was identical except for the equipment used: VR and audio-based guidance. The meditation practice lasted 10 min and involved the use of a bathing scenario in which the guided voice prompted the listener to pay attention to their breath and experience the present moment without any kind of judgment. The group of subjects assigned to the control group listened to an audiobook for the same amount of time as the meditation audio while listening to an extract from JRR Tolkein’s *The Hobbit* audiobook.

After one single meditation session, individuals undergoing the VR-based as well as audio-based interventions reported a greater level of mindfulness compared to the control intervention.

Kaplan-Rakowski et al., 2021

This study examined the effectiveness of meditation provided through VR versus meditation practice performed through video viewing and how these treatments impact the anxiety levels.

The total sample consisted of 61 business students. Women represented 57% of the sample (n = 35) and males 43% (n = 26). Individuals were randomly assigned to two groups: mediation experience mediated by VR (n = 31) or by video viewing (n = 30). The two groups had comparable levels of general self-reported anxiety. Both interventions used a scenario with animated, slow-paced visualizations of forest scenes. The VR meditation took place using an Oculus Go headset using a free app called Guided Meditation VR (http://guidedvr.com/ (accessed on 30 May 2022)). The control group watched the same video projected on a 17-inch desktop computer monitor. The meditation sessions lasted 15 min. The results showed that students in the VR group benefited from the relaxation technique to a greater extent than in the control group.

Chandrasiri et al., 2020

This is one of the few studies comparing the effects of mindfulness interventions mediated by VR with respect to a conventional awareness practice *per se*. The total sample consisted of 32 young, healthy individuals (16 males, mean age: 27.2 years) randomly assigned to either a VR (n = 16) or non-VR (n = 16) condition.

The mindfulness training exercises lasted for 20 min for both groups. In the control group, (non-VR) individuals listened to the audio track and were asked to close their eyes during the exercise or to direct their gaze down or to a single point. The experimental group listened to the same audio track while immersed in the 360 ° video/VR experience “A Walk on the Beach”.

Using the Toronto Mindfulness Scale (TMS), the authors described significant improvements in total mindfulness in both groups with respect to the pre-test phase, but no differences were found among groups. In other words, the predicted increase in awareness beyond traditional practice in VR groups was not confirmed.

Crescentini et al., 2016

This is the only study applying a mindfulness intervention over a long period (8 weeks). In fact, Crescentini et al. [47] sought to evaluate the effectiveness of the MOM program on the psychological (anxiety and awareness measures) and physiological outcomes (Heart rate and blood volume pulse amplitude) when individuals were exposed to four immersive virtual environments designed to cause different levels of stress.

The total sample consisted of 41 middle-aged healthy individuals. Twenty-one were enrolled in the experimental group (mean age: 43.3 years), whereas twenty (mean age = 36.75) were in the control group, in which participants were not involved in any meditation practice. Only the experimental group underwent a MOM course based on the Mindfulness-Based Stress Reduction program, Kabat-Zinn, [16] consisting of 8 weekly meetings of about 2 h each. Each meeting was divided into three phases: (a) 30 min of teaching on meditative practice, (b) 30 min of MOM practice divided into three exercises lasting 10 min each; and (c) debriefing for up to 1 h. At the end of the MOM training course, both groups were subjected to visually immersive scenarios aimed at eliciting different stress levels. The low-stress scenarios delineate a normal, daily life situation, while the high-stress scenarios portray an emergency. Physiological measures consisted of cardiovascular activity recorded through a photoplethysmograph and facial electromyography recorded through two sets of Ag/AgCl electrodes.

Crescentini et al. [47] demonstrated that continued meditative practice over a two-month period led to an increase in awareness skills and improved emotional regulation during stressful immersive virtual experiences. At a physiological level, heart variability and blood volume pulse amplitude were the only parameters showing significant differences in the experimental group with respect to controls. At a psychological level, the authors found more relevant findings among groups comparing FMI, MAAS, and FFMQ performance measured after training.

## 6. Discussion

In non-clinical populations, mindfulness-based interventions mediated by VR improve negative mood states (including anxiety, anger, depression, and tension) as well as mindfulness skills in non-clinical populations. Although most works included in this review describe moderate effects on psychological well-being, this field of study is characterized by several limitations:(a)The prevalent use of brief intervention (one single day lasting 5–20 min);(b)Poor use of RCT study design;(c)Poor evaluation of the impact of demographic data (work employment, gender, age) on psychological and physical outcomes;(d)High variability in technological devices and VR procedures;(e)The lack of physiological measurements to better assess physiological changes associated with mindfulness interventions, such as heart rate variability.

### 6.1. Psychological and Physiological Outcomes

Decreased anxiety seems to be the main psychological outcome of mindfulness-based interventions mediated by VR. This specific emotional improvement was reported by Crescentini et al. [47]; Tarrant et al. [46]; Kaplan-Rakowski et al. [43], using both nature-based VR scenarios as well as MOM training in train stations. The effect could extend to other negative mood states such as Anger, Depression, and Tension, as suggested by Tarrant et al. [41]. However, this kind of psychological benefit may be dependent upon a general state of relaxation and lower fatigue induced by experiencing very pleasant virtual scenarios, as suggested by Waller et al. [42].

Second, the vast majority of the reviewed studies were interested in evaluating to what extent VR impacts awareness, mindfulness skills, and state. Significant benefits were reported by Waller et al. [42] using the meditation breath attention scores and the Meditative Experiences Questionnaire; Crescentini et al. [47] using the Mindful Attention Awareness Scale and by Yildirim et al., 2021 using the State MildFulness Scale. On the other hand, Chandrasiri et al. [44] found no significant changes in the mindfulness state when an experimental VR-mediated group was compared with individuals who underwent an intervention using a mindfulness audiotrack. These similar effects induced by visual and audio-guided stimulation were also revealed by Yildirim et al. [45], who significantly reported a greater level of mindfulness state in both groups undergoing meditation practice (using audio or visual stimuli) with respect to a control group, whose individuals just listened to an audiobook. The experimental design employed by Yildirim et al. [45] is one of the more robust methodologies. In fact, to rule out the possibility that engaging in a mindfulness practice was sufficient to induce a state of mindfulness, they compared the psychological changes in two experimental groups who experienced mindfulness-based interventions with visual or audio stimulation with respect to a control group who simply listened to an audiobook. Considering this evidence together with those provided by Chandrasiri et al. [44] and considering the experimental design found in other studies, it is likely that the reported psychological benefits of mindfulness interventions are specifically related to the guidance of the mindful practice per se, irrespectively of the from sensorial stimulation.

To disentangle this critical issue, it is essential to evaluate the physiological changes associated with different mindfulness practices. Unfortunately, only Tarrant et al. [46] and Crescentini et al. [47] incorporated neurophysiological measurements in their experimental procedures. Using an EEG, Tarrant et al. [46] found that VR intervention resulted in increasing Alpha power and reducing broadband Beta activity in the anterior cingulate cortex. Investigating heart rate variability, skin conductance, respiratory frequency, and facial electromyography, Crescentini et al. [47] found that only heart rate and activity in corrugator supercilii muscle were significantly reduced by the VR intervention.

### 6.2. VR-Based Technological Considerations

Mindfulness-based interventions were carried out using specific commercial tools (Table 1). Generally, mobile VR is the easiest to use compared to tethered and standalone systems. However, the VR systems reported in Table 1 have different features and performance. Generally, tethered systems, which require expertise and a PC connection, ensure a better immersive experience compared to other VR systems, while the standalone—no connection required system—represents a compromise between affordability and an authentic, immersive VR experience. When compared to the above-mentioned VR systems, the mobile VR (supported by smartphones) is the easiest to use but performs the worst.

Oculus Rift, also known as Oculus CV1, requires an external device (PC) to work. The immersive experience in the virtual environment (VE) is ensured by a 110° of the field of view (FOV), 90 Hz refresh rate and a graphics display resolution of 2560 × 1440. Furthermore, the rotational and positional tracking also has six degrees of freedom (DOF), and the integrated headphones reproduce real-time 3D audio effects. The previous system, Oculus Rift DK2, which updated the DK1, guarantees high performance with a resolution of 960x1080 per eye, a refresh rate of 75 Hz, and 100° of FOV. In contrast, the Oculus Go VR headset [41,43] is a standalone immersive VR system. It has a fluid display (60–72 Hz) with minor FOV to 101°, and a resolution of 1280x1440. The system includes 3 DOF, a built-in sound system, and a controller that does not track position systems.

The HTC Vive [45] is an expensive, tethered, but excellent VR system. The higher immersion level is ensured by a resolution of 2440x2440 (per eye), refresh rate of 120Hz, and 6 DOF. In addition, the audio system uses on-ear headphones aligned with the ears.

The Gear VR powered by the S7 phone [46] is a mobile VR tool with accelerometer, gyroscope, and proximity sensors, and the FOV is 96°. It has the same features as the Gear VR powered by S10+ [42] with the addition of a controller and a FOV of 101°.

The HMZ-T1 of Sony [47] is a 3D viewer with built-in stereo headphones. The Head Mounted Display (HDM) connected to a PC has a 45° FOV and a resolution of 1280 × 720. The device is not equipped with sensors able to track the head movement and/or update the view in the VE. To simulate the immersion, it connects to an external sensor (InertiaCube3) with an accelerometer, gyroscope, and magnetometer. It has a refresh rate of 180 Hz of rate and 3 DOF.

### 6.3. Limitations

Nowadays, the application of mindfulness-based interventions mediated by VR in non-clinical populations has several limitations, which could explain the high variability in the results.

(1)The different duration of the mindfulness-based interventions. Most of the reviewed studies evaluated the effects of this kind of intervention after a single mindfulness exercise for a short period of time (5–20 min). Only Crescentini et al. [47] set the experimental design using an intensive long-term treatment (8 weeks). This is an important limitation of this field of study because any kind of behavioral intervention needs more time to influence brain activity, induce long-term neural modulation, and, finally, promote consistent behavioral changes.(2)Poor employment of the RCT study design. Few studies in this field have used a rigorous RCT study design. Moreover, in the few studies reported in this review, only one paper/two papers used a very active control condition where it was possible to disentangle the real effect of VR-mediated interventions with respect to other relaxation practices.(3)Poor generalizability of findings. This is mainly dependent on the enrollment of young students, with unequal numbers of each sex without information on the impact of other demographic factors (such as education) that could affect the mindfulness-based interventions.(4)The lack of consensus regarding the assessment of psychological outcomes. As shown in Table 1, to assess the primary outcome of mindfulness intervention (mood status and mindfulness skills), a plethora of behavioral tests were employed. From the perspective of translating this kind of treatment to clinical patients (i.e., anxiety disorders, cardiovascular diseases, neurological patients), the lack of a clear and established behavioral battery could mask the detection of significant psychological benefits.(5)The lack of physiological measurements to define in a more objective way changes associated with mindfulness interventions. As stated by Crescentini et al. [47], heart rate variability is strictly related to emotional activation [49], and heart rate acceleration varies consistently with stimulus arousal, increasing with both pleasant and unpleasant stimuli. Similarly, an increase or decrease in blood volume pulse amplitude has been demonstrated to be related to vasoconstriction mechanisms underlying the state of relaxation [50].

## 7. Conclusions

The application of mindfulness-based interventions in non-clinical populations is in its relative infancy since the vast majority of studies are characterized by several limitations. With this in mind, we believe that its translation to clinical populations is too premature. In defining the best practice guidelines for future studies, it is strongly recommended to:(a)Use RCTs designs with three arms, i.e., two experimental groups engaged in two different sensorial guided meditation experiences, compared with one active control condition);(b)Enroll healthy subjects of different ages, genders, and educational backgrounds, in order to evaluate the impact of demographic factors;(c)Use intensive long-term treatments to evaluate consistent behavioral and physiological changes. A follow-up re-evaluation after 3–6 months after the end of treatment will also help to reveal the persistent beneficial effects over time;(d)Define an internationally validated behavioral battery;(e)Incorporate neurophysiological measurements (ECG, EEG, and EMG) into the experimental procedure.

## Figures and Tables

**Figure 1 healthcare-10-01220-f001:**
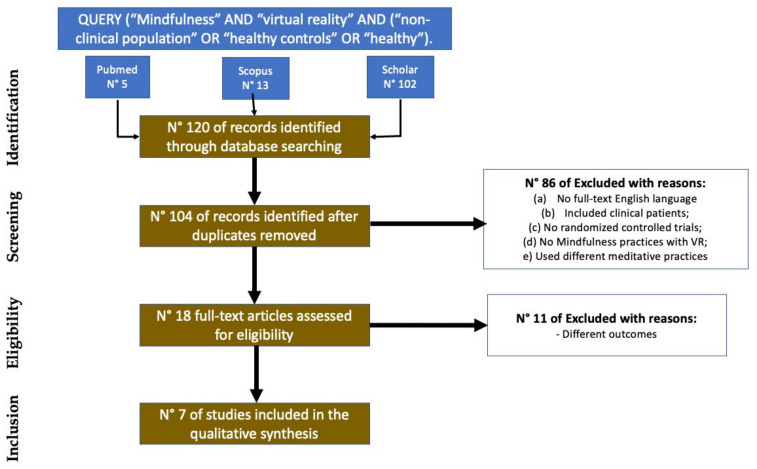
The PRISMA analysis.

**Table 1 healthcare-10-01220-t001:** Characteristics of studies applying mindfulness-based interventions with virtual reality systems.

Reference	VR System	Subjects	Mindfulness Treatment	Experimental Procedures	Outcomes	Main Results
Tarrant et al., 2022 [41]	Oculus Go VR headset	100 participants Experimental group (n°50) Control group (n°50)	During the audio-guided meditation, participants underwent a progressive body-scan/relaxation mindfulness experience.	One single meditation experience lasting 5 min in a dedicated research room, performed in two groups: The VR experience was “relaxation beach” from the Healium platform (Columbia, MO, United States) The control condition involved the simple listening of an audio track through ear headphones	The Brunel Mood Scale EEG (BioFeedback)	Both treatments led to a significant decrease in certain negative mood states (Anger, Depression, Tension). The VR group showed an increase in Happiness and Calmness scores and decreased Confusion scores.
Waller et al., 2021 [42]	Samsung Gear Headset powered by a Samsung Galaxy 10+ smartphone	82 healthy subjects Experimental group (n°41) Control group (n°41)	Individuals underwent instructor-guided meditation practices via three methods: (1) traditional face-to-face (in vivo method), (2) pre-recorded 360° video viewed by standard laptop computer monitor (2D format), and (3) pre-recorded 360° video viewed through a VR	One single meditation experience lasting 5 minutes in a dedicated research room. Participants were randomly assigned to complete two counterbalanced VR or non-VR guided meditation sessions. All participants took part in all phases of experimentation by completing a part of a 360° guided meditation pre-recorded using an HDM VR, and each participant completed a non-VR version of the same guided meditation.	LES LEC-5 ACE PCL-5 TRASC mDES BASS MEQ MBAS	VR meditation was associated with a heightened experience of awe overall with respect to other meditations. VR meditations were associated with greater experiences of relaxation (MEQ), less distractibility from the process of breathing (MBAS), and less fatigue (MEQ) compared to the 2D format.
Kaplan-Rakowski et al., 2021 [43]	Oculus Go headset	61 healthy subjects Experimental group (N = 31) VR treatment group Control group (n°30) video viewing	During the meditation, the participants underwent nature-based VR and video experiences of forest scenes, produced by Cubicle Ninjas;	One single meditation experience lasting 15 min in a classroom: Experimental group: VR-based meditation Control group: video-based meditation	Self-report Anxiety level	VR-based meditation reduced the anxiety level to a greater extent than video-based meditation.
Chandrasiri et al., 2020 [44]	Oculus Rift head-mounted display	32 healthy subjects Experimental group (N = 16) VR treatment group Control group (n°16) mindfulness audiotrack	During the audio-guided meditation, participants underwent nature-based VR experiences to immerse in “A Walk on the Beach” by Gromala et al. (2015)	One single meditation experience lasting 20 min: Experimental group: VR-based meditation Control group: mindfulness audiotrack	TMS	VR was not significantly more effective in facilitating mindfulness overall, although the VR condition was characterized by a significantly greater increase in decentering.
Yildirim et al., 2020 [45]	HTC Vive	45 healthy subjects 1° Experimental group (n°15) VR-based mindfulness intervention 2° Experimental group (n°15) audio-based mindfulness intervention Control group (n°12) listening to audiobook	During the audio-guided meditation, participants underwent nature-based VR experiences of a beach environment in which participants could experience Costa Del Sol, produced by Cubicle Ninjas; https://guidedmeditationvr.com (accessed on 30 May 2022)	One single meditation experience lasting 10 min in a dedicated research room, performed in three groups: 1° Experimental group: mindfulness VR-based experiences 2° Experimental group: mindfulness audio-based experiences Control group: Individuals listened to the first chapter of JRR Tolkien’s The Hobbit	State Mindfulness Scale Sustained Attention to Response Task	Both VR-based and audio-based interventions induced a greater level of state mindfulness, when compared to an active control intervention.
Tarrant et al., 2018 [46]	Gear VR powered by a Samsung Android s7 phone	26 healthy subjects Experimental group (n°14) Control group (n°12)	During the audio-guided meditation, participants underwent nature-based VR experiences produced by StoryUp VR (Columbia, MO, United States) using 360° video photography	One single meditation experience lasting 5 min in a dedicated research room, performed in two groups: Experimental group: non-therapeutic mindfulness VR experience Control group: The control condition was a simple 5 min eyes-open rest	STAI; GAD-7; EEG	Alpha and Beta sub-bands demonstrated slightly higher power increases on average, specifically after the VR intervention as opposed to rest.Significant reduction in Anxiety after VR was detected
Crescentini et al., 2016 [47]	Sony HMZ-T1 display	41 healthy subjects Experimental group (N = 21) VR treatment Control group(n°20)	MOM training is characterized by 2 VR experiences immersed in: (a) train station and seven tracks or a multi-floor school building.	8 weekly meetings of about 2 h each. VR immersion in POS contents VR immersion in low level of eliced stress contents VR immersion in high level of eliced stress contents	FFMQ; FMI; STAI; MAAS Cardiovascular activity	MOM led to increased mindfulness skills and reduced state and trait anxiety, as well as to better physiological and emotional regulation during low and high elicited stress experiences.

ACE—Adverse Childhood Experiences Questionnaire; BASS—Buddhist Affective States Scale; FFMQ—Five Facet Mindfulness Questionnaire; FMI—Freiburg Mindfulness Inventory; GAD-7—General Anxiety Disorder-7; ITC-SCOPI—Independent Television Company SOP Inventory; LEC-5—Life Events Checklist for DSM-5; LES—Life Experiences Survey; MAAS—Mindful Attention Awareness Scale; MBAS—meditation breath attention scores; mDES—Modified Differential Emotions Scale; MEQ—Meditative Experiences Questionnaire; MOM—mindfulness-oriented meditation; PCL-5—Posttraumatic Stress Disorder Checklist for DSM-5; SSQ—Simulator Sickness Questionnaire; STAI—State-Trait Anxiety Inventory; TMS—Toronto Mindfulness Scale; TRASC—Trauma-Related Altered States of Consciousness Items; VAS—Visual analog scale; VR—virtual reality.

## Data Availability

Not applicable.

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
