# Peer review of "Mediating Mindfulness-Based Interventions with Virtual Reality in Non-Clinical Populations: The State-of-the-Art"

_healthcare, 2022, doi:10.3390/healthcare10071220_

Round 1

Reviewer 1 Report

This narrative review provides the state-of-art mindfulness interventions mediated by VR systems in non-clinical populations to evaluate its real effectiveness in influencing mood and physiological status. Overall, the main recent scientific papers summarizing evidence is convincing. However, I suggest that the authors should revise their English writing, since many sentences are very long, which affects readability.

Author Response

REVIEWER n°1

This narrative review provides the state-of-art mindfulness interventions mediated by VR systems in non-clinical populations to evaluate its real effectiveness in influencing mood and physiological status. Overall, the main recent scientific papers summarizing evidence is convincing. However, I suggest that the authors should revise their English writing, since many sentences are very long, which affects readability.

REPLY: we would like to thank this reviewer for these positive comments. The entire paper has been edited and proofread by an English writer. Please check the E4AC manuscript declaration.

Reviewer 2 Report

Mindfulness is one of the most popular psychotherapeutic techniques assisting people in in promoting good mental and physical health. When combined with immersive virtual reality (VR) it has been proven to be especially effective for a wide range of mood disorders where traditional mindfulness is currently proving valuable. However, the vast majority of immersive VR-enhanced mindfulness applications have been dedicated to clinical settings, whereas there is a paucity of evidence on healthy subjects.

The authors proposed a  narrative review aimed at providing state-of-art mindfulness interventions mediated by VR systems in non-clinical populations in order to evaluate its real effectiveness in influencing mood and physiological status.

The authors focused to researches  with an RCT study design.

They highlighted   that: (a)  the vast majority of studies are characterized by one single meditation experience which would seem enough to induce a significant reduction in negative mood states (anxiety, anger, depression, tension) combined with increased mindfulness skills. (b) Instead, physiological correlates of mindfulness practices have sparsely been investigated. (c) The application of VR-enhanced mindfulness-based interventions in non-clinical populations is in its relative infancy since the vast majority of studies are characterized by several weaknesses, such as the poor employment of RCT study design, the lack of physiological measurements (i.e., heart rate variability), as well as by the high heterogeneity in demographical data, technological devices and VR procedures.

They concluded that  before applying mindfulness interventions mediated by VR in clinical populations there is a need to define more robust and reliable methodological procedures.

The manuscript is interesting. However it should be rearranged according to the standards as the readers expect a standard structure for the review.

These are my comments:

1.      Rearrange the manuscript according to the standards following the further comments. I understand that  this is not a systematic review, however it must show a clear design. It is not clear where the review starts. It seems that the review starts from references [38] in section 3. I suggest to rearrange the review into points of views. The first two sections become the results of the first two points of view.  The section 3, becomes the output of the third point of view using the RCTs.

2.      Insert a brief introduction with the hypothesis, problems, etc.

3.      Insert a clear purpose with the key questions the review must answer.

4.      A short methods section must describe the strategy of the review, databases searched, keys and synthesis. I think it differs for the three points of view.

5.      Usually the conclusions do not have references

6.      Delete “Vecchio elenco di bibliografia”

In summary I like this contribute, it sounds scientifically, and it is useful to the scholars.  I’d like to see it again with a strong revision in order to be better appreciated in the scientific community.

Author Response

REVIEWER n°2

Mindfulness is one of the most popular psychotherapeutic techniques assisting people in in promoting good mental and physical health. When combined with immersive virtual reality (VR) it has been proven to be especially effective for a wide range of mood disorders where traditional mindfulness is currently proving valuable. However, the vast majority of immersive VR-enhanced mindfulness applications have been dedicated to clinical settings, whereas there is a paucity of evidence on healthy subjects.

The authors proposed a  narrative review aimed at providing state-of-art mindfulness interventions mediated by VR systems in non-clinical populations in order to evaluate its real effectiveness in influencing mood and physiological status. The authors focused to researches  with an RCT study design.  They highlighted  that: (a)  the vast majority of studies are characterized by one single meditation experience which would seem enough to induce a significant reduction in negative mood states (anxiety, anger, depression, tension) combined with increased mindfulness skills. (b) Instead, physiological correlates of mindfulness practices have sparsely been investigated. (c) The application of VR-enhanced mindfulness-based interventions in non-clinical populations is in its relative infancy since the vast majority of studies are characterized by several weaknesses, such as the poor employment of RCT study design, the lack of physiological measurements (i.e., heart rate variability), as well as by the high heterogeneity in demographical data, technological devices and VR procedures.  They concluded that  before applying mindfulness interventions mediated by VR in clinical populations there is a need to define more robust and reliable methodological procedures.

The manuscript is interesting. However it should be rearranged according to the standards as the readers expect a standard structure for the review.

REPLY: We would like to thank this reviewer for these positive comments. We feel that our manuscript is strongly improved by incorporating their suggestions. In the attached reply to the reviewer, we outlined our responses to each of their comments.

These are my comments:

  1. Rearrange the manuscript according to the standards following the further comments. I understand that this is not a systematic review, however it must show a clear design. It is not clear where the review starts. It seems that the review starts from references [38] in section 3. I suggest to rearrange the review into points of views. The first two sections become the results of the first two points of view.  The section 3, becomes the output of the third point of view using the RCTs.

REPLY: We would like to thank this reviewer for this suggestion. The manuscript has been rearranged in accordance with Preferred Reporting Items for Systematic Reviews and Meta-Analyses (PRISMA) guidelines, in accordance with the suggestions made by reviewer n°3.

  1. Insert a brief introduction with the hypothesis, problems, etc.

REPLY: Done

  1. Insert a clear purpose with the key questions the review must answer.

REPLY: Done

  1. A short methods section must describe the strategy of the review, databases searched, keys and synthesis. I think it differs for the three points of view.

REPLY: See the new methods section

  1. Usually the conclusions do not have references

REPLY: Removed

  1. Delete “Vecchio elenco di bibliografia”

REPLY: Removed

In summary I like this contribute, it sounds scientifically, and it is useful to the scholars.  I’d like to see it again with a strong revision in order to be better appreciated in the scientific community.

Reviewer 3 Report

The study is interesting. However, several questions can be raised to the methods related to the selection of the papers.

Currently, several well-established methods are available to enable a systematic approach when selected the studies in review studies.

It is necessary to know in detail how the selection process have been conducted. For example, how many databases have been screened, what is the search query applied in those databases, what are the inclusion and  exclusion criteria and so on…

I recommend the authors to use PRISMA method.

Author Response

REVIEWER n°3

The study is interesting. However, several questions can be raised to the methods related to the selection of the papers.

Currently, several well-established methods are available to enable a systematic approach when selected the studies in review studies.

It is necessary to know in detail how the selection process have been conducted. For example, how many databases have been screened, what is the search query applied in those databases, what are the inclusion and  exclusion criteria and so on…

I recommend the authors to use PRISMA method.

 REPLY: We would like to thank this reviewer for this suggestion. The manuscript has been rearranged in accordance with Preferred Reporting Items for Systematic Reviews and Meta-Analyses (PRISMA) guidelines, in accordance with the suggestions made by reviewer n°2. 

Round 2

Reviewer 2 Report

The authors addressed all the comments.

The manuscript improved a lot.

There are not further comments

Author Response

Dear reviewer thanks a lot for your final evaluation

Reviewer 3 Report

The screening process is not clear. Particularly, when the authors show 54456 identified studies and only 104 have been screened according to abstract and title.

Moreover, the research questions are not provided.

The main contribution of a literature survey is to synthesize the existing body of knowledge in a unique way which the readers won't get by simply reading the cited papers alone. The goal is to identify common threads and gaps that would open up new challenging, interesting and significant research directions.

In addition to attempting to provide a comparison between different techniques, make sure this is not limited and the dimensions against which this is attempted are adequate and offer valuable or significant insights to the research domain as a whole.

Although this is attempted to some extent, additional reflection and exploration of the issues involved should be included so as to offer more valuable insights to the reader.

Author Response

The screening process is not clear. Particularly, when the authors show 54456 identified studies and only 104 have been screened according to abstract and title. Moreover, the research questions are not provided.

REPLY: we would like to thank this reviewer for this suggestion. PRISMA analysis and description have been now re-formulated.

The main contribution of a literature survey is to synthesize the existing body of knowledge in a unique way which the readers won't get by simply reading the cited papers alone. The goal is to identify common threads and gaps that would open up new challenging, interesting and significant research directions. In addition to attempting to provide a comparison between different techniques, make sure this is not limited and the dimensions against which this is attempted are adequate and offer valuable or significant insights to the research domain as a whole. Although this is attempted to some extent, additional reflection and exploration of the issues involved should be included so as to offer more valuable insights to the reader.

REPLY: To the best of our knowledge, we provide a literature survey that has the merit to highlight that there is a paucity of rigorous methodological study designs where the effectiveness of a specific behavioral treatment ("immersive VR-enhanced mindfulness practices") has been widely reported using one-single session, with different VR systems, with different psychological batteries, with different VR-related scenarios and (last but not least) with a poor employment a control group.

For this reason, we believe that this NARRATIVE review will contribute to the advancement of this field of study where there is an urgency to define a specific, clear, robust and reliable methodology in the healthy population before applying "immersive VR-enhanced mindfulness practices" in clinical settings.